# Relative Contribution among Physical Fitness Factors Contributing to the Performance of Modern Pentathlon

**DOI:** 10.3390/ijerph18094880

**Published:** 2021-05-03

**Authors:** Byoung-Goo Ko, Eun-Hyung Cho, Jin-Seok Chae, Ji-Hoon Lee

**Affiliations:** 1Korea Institute of Sport Science, Seoul 01794, Korea; bgko@kspo.or.kr (B.-G.K.); ehcho@kspo.or.kr (E.-H.C.); 2Department of Sport Science, Yongin University, Yongin 17092, Korea; chejinseok0511@hanmail.net; 3Daegu Gyongbuk Institute of Science Technology (DGIST), Daegu 42988, Korea

**Keywords:** elite fencers, fencing specific training, physical fitness, performance

## Abstract

This study reveals the relationship between physical fitness factors and performance in modern pentathlon and identifies the contribution of each physical factor to overall performance. The physical fitness assessment data and the competition records collected by the Korean national team pentathletes for the years 2005 to 2019 were tracked. The correlation between the competition records and fitness factors was confirmed by correlation analysis. In addition, the physical factors affecting performance were identified through multiple regression analysis, and the average difference between national and international competitions was verified by *t*-test. The first result was that fencing, swimming, and horseback riding rankings were more relevant to the overall pentathlon performance score than the combined rankings in national competitions. In the international competitions, performance in the combined running and shooting event was more relevant than fencing, swimming, and horseback riding. Second, the basic fitness factors of grip strength and sergeant jump and the specific fitness factors of leg strength—left and right average flexor were correlated with overall pentathlon performance national competitions. However, in international competitions, sergeant jump, 20 m shuttle run, reaction time, lung capacity, and back strength were correlated (presented in high to low order). In terms of the specific fitness factors, relative (%BW) and absolute (Nm) leg strength—left and right average flexor, lower body anaerobic fatigue rate, half squat, relative (W/kg) and absolute (Watts) maximal lower body anaerobic power were correlated accordingly with overall pentathlon performance. Third, we analyzed the differences between average performance in national and international competitions. Only the combined running and shooting event out of the five modern pentathlon events showed a difference. Grip strength and relative leg strength—average extensor AP (%BW) appeared to be different among the physical fitness factors. Fourth, we examined the level of contribution of each of the fitness factors on overall performance. The model’s goodness of fit was confirmed, and grip strength was found to have a significant contribution on overall performance. Furthermore, the level of contribution was higher in the following order: relative leg strength—left and right average flexor (%BW), bench press, half squat, relative leg strength—average extensor AP(%BW), GXT—time to exhaustion, relative lower body anaerobic average power (W/kg), and maximal lactic acid concentration. With the 2020 Tokyo Olympics just around the corner, combined running and shooting performance appeared to be a decisive factor in the final ranking in modern pentathlon according to the analysis of the basic and specific fitness factors of pentathletes. The basic fitness factors are critical in order of sergeant jump, grip strength, reaction time, lung capacity, side-step, back strength, 20 m shuttle run, sit-and-reach, sit-ups, and single leg standing. With respect to the specific fitness factors, relative leg strength—left and right average flexor (%BW), bench press, half squat, relative leg strength—average extensor AP (%BW), GXT—time to exhaustion, relative lower body anaerobic average power (W/kg), and maximal lactic acid concentration showed their relevance accordingly.

## 1. Introduction

The modern pentathlon comprises five events, which are fencing (épée), swimming, horseback riding (obstacle race), and a combined event of shooting and cross country running. The cumulative point standings from the first three events (fencing, swimming, and horseback riding) decide the competitors’ order at the start line for the cross country running and shooting combined event. Thus, a unique aspect of modern pentathlon is that its point system plays an important role in the competition between individuals. Pentathletes are ranked based on their cumulative points from five events on a single day: fencing (épée: 1 min round-robin), swimming (200 m), horseback riding (350 m obstacle course), and combined (pistol shooting and cross country running 3200 m) [1]. Regarding the difficulty of each event, simultaneously competing in all five events requires pentathletes to possess traits of robust stamina, concentration, and skilled techniques. Specifically, the level of stamina is closely related to the performance of elite pentathletes. Stamina is composed of multiple factors. By studying the factors affecting stamina and their influence on pentathlete performance, along with using valid assessment tools and structured training, successful achievement at international competitions can be achieved [2,3].

In a study of elite pentathletes’ physical characteristics and sport-related factors, Jung and his colleagues [4] reported no significant difference in anthropometric data, despite national team athletes showing statistically higher scores on grip strength, 2000 m run, standing long jump, and sit-ups variables compared with varsity athletes. Therefore, maintenance of a high level of strength, flexibility, cardiorespiratory fitness, and agility was suggested in order to improve performance in modern pentathlon. Choi’s study [5] of 21 elite pentathletes and performance factors illustrated that reaction time and Harvard step test number were related to running performance and that grip strength was related to shooting performance. Additionally, agility (standing long jump) was reported to be related to fencing, but not to swimming and horseback riding. With respect to regression analysis, the factors affecting cross country running were side-step, backward flexibility, single leg standing, and Harvard step test. Factors affecting swimming performance were back muscle strength, pull-ups, and Harvard step test. Grip strength was shown to affect the shooting event, while back muscle strength, pull-ups, standing long jump, trunk forward flexion, single leg standing, and Harvard step test score were reported as factors affecting fencing performance. However, horseback riding performance factors were reported as being unclear.

Meanwhile, Lim et al. [6], by examining performance factors on 14 pentathletes, noted that there was no significant difference in shooting performance between elite and non-elite level pentathletes, but that there was a significant difference in their cross country running. Furthermore, Kim et al. [7] presented significant differences in sectional running time, speed, shooting time, and transition between superb and ordinary athletes by studying 16 pentathletes’ performances on different skills, suggesting that power and endurance development improves running performance.

Um [8] investigated the relationship between pentathlon’s shooting, cross country running, and combined event performances, showing a correlation between the running score under the previous pentathlon regulations and the newer combined score, but no significant correlation between the shooting score under the previous regulations and the newer combined score. Therefore, they suggested that running is the critical performance factor if a pentathlete has intermediate or higher shooting skills. Kim [9] emphasized the importance of cardiorespiratory fitness and fatigue recovery in his study on exercise prescription for cardiorespiratory fitness in national team pentathletes. In a study on physiological factors in pentathletes, Yoon et al. [10] presented that the Korean national team pentathletes had significantly lower VO_2max_ and maximal ventilation capacity compared with world class pentathletes, proposing the importance of interval training for cardiorespiratory fitness and lactic acid tolerance in order to improve their game performances.

Moreover, although numerous efforts were made to understand the performance factors in pentathlon, like Kim et al. [11], who examined different types of techniques in pentathlon fencing, and Kang et al. [12], who compared physical fitness of decathletes and pentathletes, there are still a lot of unexplained aspects due to the small number of subjects in the studies and the questionable validity of assessment measures that have been used.

Because the COVID-19 pandemic postponed the 2020 Tokyo Olympics, pentathletes have had extra time to train and prepare for competition. Consequently, research that considers pentathletes’ performance on different events in the competition and their relationship with physical fitness factors is timely and valuable [13].

Thus, this study aimed to reveal the relationship between fitness factors and pentathlon performance, compare national and international competitions, and explain the relative contribution of each fitness factor to different pentathlon events [14]. This study provides valid evidence for fitness assessment and testing selection, and can also be utilized in the process of pentathlete selection, fitness assessment, training distribution and planning because it includes objective investigation of the pentathlon performance factors.

## 2. Materials and Methods

### 2.1. Subjects and Measured Variables

The research subjects were male pentathletes registered in a performance assessment system who experienced a fitness assessment by our institution between 2005 and 2019 by being chosen for the Korean national team. The data include their fitness assessments and records from national (*n* = 57) and international (*n* = 57) competitions. The competitions records were classified into national and international. The records were quantified and were collected from the Korean Modern Pentathlon Federation and Union Internationale de Pentathlon Moderne (UIPM) websites. Additionally, we obtained fitness assessment data on each year’s national team pentathletes from a performance assessment system.

### 2.2. Data Collection

The national competition records (*n* = 57) were collected from the Korean Modern Pentathlon Federation website and included 4 different competitions (National Sports Festival; National Pentathlon; Minister of Culture, Sports, and Tourism Pentathlon; and …). The international competition records (*n* = 57) for the competitions in which the subjects participated were obtained from the UIPM website (http://www.uipmworld.org, accessed on 5 May 2020). The physical fitness and anthropometric data were retrieved from the Performance Assessment System (http://nkeft.sports.re.kr, accessed on 10 June 2020), which examined the physical fitness (basic and specialized) of the national team pentathletes each year.

### 2.3. Data Analysis and Procedure

This study compared pentathlon performance factors between the international and national competitions by independent *t*-test and examined with multiple regression analysis the relative contribution of basic and specialized physical fitness variables to modern pentathlon game performance. Because the pentathlon point system was revisited, the former competition records were normalized. Collected data were organized with Microsoft Excel version 2016 and analyzed with SPSS (IBM, Chicago, USA) version 27.0. detailed procedure and methods are as follows. First, we selected the national team male pentathletes who were assessed by KISS (Korea Institute of Sport Science) between 2005 and 2019 and collected each year’s national team’s physical fitness data, which were registered in the Performance Assessment System. Second, we retrieved and quantified the data from the Korean Modern Pentathlon Federation and Union Internationale de Pentathlon Moderne (UIPM) websites from 2005 to 2019. Third, we examined the relationship between physical fitness and modern pentathlon performance factors (including 5 sub-events) by correlation analysis. Fourth, we identified the relative contribution of the basic and specific physical fitness variables to overall pentathlon performance [15].

## 3. Results

### 3.1. Relationship between Physical Fitness, Anthropometric Data, and Modern Pentathlon Performance

We examined the correlation between national and international competition records. The result of the correlation analysis is illustrated in (Table 1). The modern pentathlon consists of a combined shooting and running event and fencing, swimming, and horseback riding events. The summation of points from those five events provides an overall performance score. The overall performance score decides each competitor’s rank. The summation of shooting and running, or the combined score had the highest correlation with the rank in modern pentathlon. Additionally, the combined score had the highest correlation with the overall performance score in both national (0.726) and international (0.956) competitions. A higher correlation was found between the overall performance score and the combined score in the international competitions, while the national competition records were more relevant to fencing, swimming, and horseback riding (Table 2).

### 3.2. Relationship between Physical Fitness, Anthropometric Data, and the Overall Performance in International and National Modern Pentathlon

#### 3.2.1. National Competition

In terms of the basic physical fitness factors, the performance in fencing showed a significant correlation with lung capacity (−0.410) and 20 m shuttle run (−0.267). When it comes to the specific physical fitness factors, leg strength—left and right average flexor (−0.394) and half squat (−0.390) were correlated. For swimming performance, grip strength (−0.441) and 20 m shuttle run—two basic fitness factors—were correlated. Among the specific fitness factors, maximal leg strength-average extensor AP (0.394) and leg strength-left and -right average flexor (−0.281) showed significant correlation. Horseback riding performance was significantly related to reaction time (0.290) and 20 m shuttle run (−0.278), while reporting no significant correlation with the specific fitness factors. The combined (running and shooting) performance was related with sergeant jump (−0.345). In the case of specific fitness factors, the combined running and shooting event performance showed a relationship with leg strength—left and right average flexor %BW (−0.371), leg strength—left and right average flexor Nm (−0.331) and maximal leg strength—average extensor AP (−0.315). Lastly, the overall performance score and the basic fitness factors appeared to have significant correlations with grip strength (−0.327) and sergeant jump (−0.327). Between the specific fitness factors, leg strength—left and right average flexor (−0.393) and leg strength- left and right average flexor (−0.339) showed a correlation(Table 3).

#### 3.2.2. International Competition

The fencing score had no correlation with the basic fitness factors, but was correlated with two specific fitness factors: lower body anaerobic—fatigue rate (−0.303) and lower body anaerobic—maximal power (−0.261). The swimming performance also had no correlation with the basic fitness factors but was correlated with two specific fitness factors: leg strength—left and right average flexor (−0.327) and GXT—VE_max_ (0.277). Furthermore, the horseback riding performance was correlated with leg strength—left and right flexor (−0.332) and GXT—HR_max_ (0.29), while showing no relationship with the basic fitness factors. The combined performance was shown to be correlated with sergeant jump (−0.430), 20 m shuttle run (−0.374), lung capacity (−0.361), reaction time (0.333), resting heart rate (−0.292), grip strength (−0.286), and back strength (−0.279). When it came to the specific fitness factors, combined performance appeared to have correlation with leg strength—left and right average flexor (−0.371), lower body anaerobic—fatigue rate (0.406), and half squat (−0.315). Lastly, the overall performance score was correlated with sergeant jump (−0.439), 20 m shuttle run (−0.350), reaction time (0.349), lung capacity (−0.323), and back strength (−0.286). For the specific fitness factors, relative leg strength—left and right average flexor (−0.500), absolute leg strength—left and right average flexor (−0.440), lower body anaerobic fatigue rate (0.433), half squat (−0.338), relative lower body anaerobic maximal power (0.315), and absolute lower body anaerobic maximal power (0.287) were correlated with the overall performance score(Table 4).

### 3.3. Comparison of the Average Overall Performance between the International and National Competition

First, on this comparison between the international and national competition, out of the five events of modern pentathlon, only the combined performance score, which includes shooting and running, appeared to have a significant difference in average overall performance. Second, differences in the anthropometric data were not significant, while the grip strength data showed a significant difference between the international and national competition. Third, in terms of the specific fitness factors, only relative maximal leg strength—average leg extensor AP (%BW) showed a significant difference (Table 5 and Table 6).

### 3.4. The Physical Fitness Factors and Their Level of Contribution to Modern Pentathlon Performance

#### 3.4.1. The Basic Fitness Factors and Their Contribution to Overall Performance

To analyze the level of contribution of the physical fitness factors on modern pentathlon performance, multiple regression analysis was used with the fitness factors as independent variables and overall performance as the dependent variable. In order to see each variable’s contribution, we utilized a collective input method for those independent variables. By analysis of variance, we confirmed the model’s goodness of fit, and the regression model proved to be significant with an *F* value of 3.102. The adjusted *R*-squared was 0.157, showing that 15.7% of the total variation was explained by the estimated regression model. By examination of the statistically significant regression coefficient, among the basic fitness factors only grip strength contributed to the predicted overall performance. Additionally, we considered the absolute value of standardized regression coefficients (*β*), which enabled evaluation of the relative contribution of each independent variable in our test of the level of contribution of the basic fitness factors to overall pentathlon performance. In order of magnitude, sergeant jump, reaction time, lung capacity, side step, back strength, sit-and-reach test, sit-up, and single leg stance contributed significantly.

#### 3.4.2. The Specific Fitness Factors and Their Contribution to the Overall Performance

The results of the multiple regression analysis setting the specific fitness factors as an independent variable and overall performance as the dependent variable appears in Table 7 and Table 8. By analysis of variance, we confirmed the model’s goodness of fit, with an *F* value of 6.451 (*p* < 0.001) showing that the regression model is significant. The adjusted *R*-squared was 0.252, showing that 25.2% of the total variation can be explained by the estimated regression model. To examine each variable’s contribution, we collectively input the independent variables. By examination of the statistically significant regression coefficients, relative leg strength—left and right average flexor, bench press, relative leg strength—average extensor, and half squat contributed to the overall performance in terms of the specific fitness factors. Additionally, we considered the absolute value of standardized regression coefficients (*β*), which enabled evaluation of the relative contribution of each independent variable in our test of the level of contribution of the specific fitness factors to the overall pentathlon performance. In the order of relative magnitude, leg strength—left and right average flexor, bench press, half squat, relative maximal leg strength—average extensor AP, GXT—exhaustion time, relative lower body anaerobic—average power, and maximum lactic acid concentration, had high contributions.

## 4. Discussion

This study examined the relationship between physical fitness factors and performance in the modern pentathlon as well as each factors’ level of contribution to performance. We collected the Korean national team pentathletes’ national and international competition records and fitness assessment data for the years 2005 to 2019 to analyze the relationship between the various factors. We found the following four results. First, among the five events in the modern pentathlon, performance in the combined shooting and running event is the most relevant predictor of overall performance [16].

However, we must take into account the rule that pentathletes are given priority positions for starting the combined event based on their scores from the first three events (fencing, swimming, and horseback riding scores) in the national competition, and also must take into account that the combined running and shooting score in the international competitions showed a high correlation with overall pentathalon performance. In other words, the combined running and shooting event was a critical factor for the final rank in the international competition, while the fencing, swimming, and horseback riding score impacted the subsequent combined running and shooting event that tended to decide the winner in the national competitions. Based on this trend that the combined (running and shooting) score leads to a pentathlete’s final rank, Lee and Ahn (2012) suggested that the Korean pentathletes are approximately 3.8% behind in shooting performance compared with the world record holder, and thus they need to be able to cover 3000 m in under 9 min and 20 s during the running segment in order to reach a world record [17]. Their claim supports this study’s finding that indicates that the combined (running and shooting) performance has a huge impact on the final rank in the international competitions [18].

Second, the fencing event, which has the highest individual variations and requires a comprehensive game control ability due to 35-round 1 vs. 1 matches, showed the highest correlation with lung capacity and 20 m shuttle run among the basic fitness factors and the highest correlation with leg strength—left and right average flexor and half squat among the specific fitness factors.

For swimming, grip strength, 20 m shuttle run, maximal leg strength—left and right average extensor AP and leg strength—left and right average flexor showed significant correlations. For HS, reaction time and 20 m shuttle run showed high correlations in terms of the basic fitness factors. Sergeant jump, relative and absolute leg strength—left and right average flexor, and maximal leg strength were found to have high correlation with CS.

In the international competitions, the FS only showed a relationship with lower body anaerobic—fatigue rate and maximal power. This can be explained by the importance of lower body fitness in fencing. Likewise, Youngsun et al. [11] presented findings showing that foreign pentathletes had superior situational reaction and quick thrust techniques compared with the Korean pentathletes, which partly corresponds to this study’s result of how lower body anaerobic—fatigue rate and maximal strength displayed high correlation with the overall performance.

Leg strength—left and right average flexor and GXT—VE_max_ were only correlated with SS. This result demonstrates that the Korean national team pentathletes are about 3.2% behind the world record holder, complementing Lee and Ahn’s study [3], which suggested that pentathletes need to score a minimum 180 out of 200 to achieve world-class performances. Therefore, pentathletes could possibly reach close to the world record by developing certain factors: GXT—VE_max_ and relative leg strength—left and right average flexor. We thus conclude that there is an important need for relative strength development.

Similar to SS, HS was also correlated with GXT—HR_max_ and relative and absolute leg strength—left and right average flexor. Other fitness factors related to HS were flexibility, lower body strength, balance, and agility. The basic fitness factors correlated with CS (running and shooting) were sergeant jump, 20 m shuttle run, lung capacity, reaction time, resting heart rate, grip strength, and back strength. In terms of the specific fitness factors, relative leg strength—left and right average flexor, lower body anaerobic fatigue rate, and half squat showed significant correlations with CS, while maximal leg strength and absolute leg extensor strength can be interpreted as having notable contributions as well. These results are thought to be grounds for achieving the conclusions of the research by Lee and Ahn [3], which stated that the Korean national pentathletes were about 3.8% behind the world record holders and needed an average of 90% accuracy in shooting and the ability to cover 3000 min 9 min 20 s in order to be closer to world-class performances [19].

The basic fitness factors correlated with the overall performance were, in order of magnitude, sergeant jump, 20 m shuttle run, reaction time, lung capacity, and back strength. With respect to the specific fitness factors, the relative order correlations was relative and absolute leg strength—left and right average flexor, lower body anaerobic fatigue rate, half squat, and relative and absolute lower body maximal anaerobic power.

Third, we examined the differences between the anthropometric data and the basic and specific fitness factors between the national and international competitions. In this sport of modern pentathlon, only the combined event displayed a difference. Moreover, grip strength in the basic fitness factors and relative leg strength—average flexor AP in the specific fitness factors appeared to be correlated with average values.

Fourth, we studied the level of contribution of the basic fitness factors on overall pentathlon performance. The only factor in the regression model (with confirmed goodness of fit) that was proved to contribute to overall performance was grip strength. Additionally, the level of contribution was highest in the order of sergeant jump, grip strength, reaction time, lung capacity, side step, back strength, 20 m shuttle run, sit-and-reach, sit-ups, and single leg standing. Among the specific fitness factors, relative leg strength- average flexor, bench press, half squat, relative leg strength—average extensor AP, GXT—time to exhaustion, relative lower body anaerobic average power, and maximal lactic acid concentration had higher contributions, accordingly.

## 5. Conclusions

This study can provide valid evidence for the utility of fitness assessment and testing selection and can also be utilized in the pentathlete selection process, fitness assessment, training distribution, and planning because it includes objective investigation of pentathlon performance factors. Based on correlation and multiple regression analysis, we concluded that muscular endurance, cardiorespiratory fitness, and agility are the critical factors in modern pentathlon game performance. According to interviews with a number of modern pentathlon experts, flexibility, upper body and shoulder muscular strength, and endurance are considered to be crucial factors for SS. In the same manner, flexibility, lower body muscular strength, balance, and dexterity are important in HS. For FS, agility, dexterity and lower body muscular strength are critical. For CS, cardiorespiratory fitness, muscular endurance, and agility are thought to play important roles. Overall, muscular endurance, cardiorespiratory fitness, and agility are decisive factors in the game of modern pentathlon.

## Figures and Tables

**Table 1 ijerph-18-04880-t001:** The correlational relationships between the national and international competition and overall and selected individual events performance in modern pentathlon.

		OP	Fencing	Swimming	Horseback Riding	Combined Running and Shooting
National competition	Fencing	0.420 **				
	Swimming	0.462 **	−0.013			
	Horseback riding	0.678 **	0.093	0.377 **		
	Combined running and shooting	0.726 **	0.307 *	0.122	0.036	
International competition	Fencing	0.244				
	Swimming	0.404 **	0.025			
	Horseback riding	0.422 **	0.115	0.283 *		
	Combined running and shooting	0.956 **	0.133	0.286 *	0.167	

* *p* < 0.05, ** *p* < 0.01, OP: Overall Performance.

**Table 2 ijerph-18-04880-t002:** Overall performance/combine/(fencing + swimming + horseback riding) Spearman‘ Rho.

Domestic Competition				International Competition			
	OP	Combined	FSH		OP	Combined	FSH
OP				OP			
Combined	0.734 **			combine	0.876 **		
FSH	0.789 **	0.204		FSH	0.639 **	0.271 *	

* *p* < 0.05, ** *p* < 0.01, Combined: running + shooting, FSH: fencing + swimming + horseback riding.

**Table 3 ijerph-18-04880-t003:** Correlation between anthropometric data and physical fitness factors and modern pentathlon and overall performance in national competitions.

Strength	Event	FS	SS	HS	CS	OP
Basic Fitness Factors	Height (cm)	0.241	−0.138	−0.078	−0.134	−0.111
	Body fat percentage (%)	0.203	−0.328 *	−0.147	−0.093	−0.164
	Weight (kg)	0.147	−0.226	−0.05	−0.065	−0.083
	BMI (kg/m^2^)	−0.035	−0.167	0.019	0.053	0.01
	Back strength (kg)	−0.239	−0.091	0.083	−0.218	−0.147
	Grip strength (kg)	−0.061	−0.441 **	−0.172	−0.226	−0.327 *
	Repeat jump (count/30 s) 45 cm	0.050	−0.312 *	−0.101	−0.249	−0.259
	Sit-up(count/60 s)	−0.240	0.233	0.188	−0.229	−0.039
	Sergeant Jump (cm)	−0.136	−0.163	−0.101	−0.345 **	−0.327 *
	Reaction time (sound) (1/1000 s)	0.008	0.188	0.290 *	0.081	0.251
	Side step (count/20 s)	−0.015	0.093	0.108	−0.092	0.017
	Lung capacity: FVC (cc/kg)	−0.410 **	−0.094	−0.096	−0.083	−0.195
	Resting heart rate (count/min)	−0.176	−0.156	−0.167	0.009	−0.148
	20 m shuttle run (count)	−0.267 *	−0.337 *	−0.278 *	−0.068	−0.307 *
	Sit-and-reach test (cm)	−0.141	−0.284 *	−0.023	0.006	−0.083
	Single leg Stance (s)	0.022	0.001	0.041	0.164	0.131
Specific Fitness Factors	Bench press (kg)	−0.267 *	0.177	0.03	−0.072	−0.045
	Half squat (kg)	−0.390 **	0.044	0.027	−0.237	−0.194
	Low body anaerobic (30 s)— average power (W/kg)	0.137	−0.011	0.089	0.057	0.109
	Low body anaerobic (30 s)— fatigue rate (%)	0.059	0.157	0.148	0.168	0.228
	Low body anaerobic (30 s)— maximum power (W/kg)	0.102	0.042	0.131	0.129	0.181
	Low body anaerobic (30 s)—average power (Watts)	0.188	−0.102	0.067	0.044	0.081
	Low body anaerobic (30 s)—maximum power (Watts)	0.196	−0.022	0.1	0.119	0.163
	Leg strength (60°/s)—Left and right average flexor (%BW)	−0.394 **	−0.241	−0.084	−0.371 **	−0.393 **
	Leg strength (60°/s)—Left and right average flexor (Nm)	−0.253	-0.281 *	−0.066	−0.331 *	−0.339 **
	Leg strength (60°/s)—Left and right average extensor (%BW)	−0.172	0.03	0.07	−0.012	0.008
	Leg strength (60°/s) and right average extensor (Nm)	−0.178	0.026	0.158	−0.177	−0.048
	GXT(Treadmill)—Drained time	−0.096	−0.017	0.081	0.082	0.079
	GXT(Treadmill)—AT (%VO_2__max_)	0.121	−0.057	−0.163	0.095	−0.022
	GXT(Treadmill)—AT (㎖/kg/min)	0.107	−0.129	−0.118	0.065	−0.03
	GXT(Treadmill)—HR@AT (count/min)	0.118	−0.1	0.037	0.189	0.144
	GXT(Treadmill)—HR_max_ (count/min)	0.060	−0.038	0.132	0.208	0.212
	GXT(Treadmill)—VE_max_ (ℓ/min)	−0.067	0.203	0.028	−0.049	0.008
	GXT(Treadmill)—VO_2__max_ (㎖/kg/min)	0.018	−0.143	0.013	−0.026	−0.03
	Maximum lactic acid (mMol)	−0.121	0.148	0.135	−0.126	0.002
	Maximum leg strength ((180°/s)—average extensor AP (%BW)	−0.047	0.394 **	0.218	−0.02	0.171
	Maximum leg strength (180°/s)—average extensor AP (watts)	−0.119	0.186	0.057	−0.315 *	−0.155

* *p* < 0.05, ** *p* < 0.01, FS: fencing score, SS: swimming score, HS: horseback riding score, CS: combined score.

**Table 4 ijerph-18-04880-t004:** Correlation between anthropometric data and physical fitness factors and modern pentathlon and overall performance in international competitions.

Strength	Event	FS	SS	HS	CS	OP
Basic fitness factors	Height (cm)	0.058	−0.136	−0.117	−0.164	−0.178
	Body fat percentage (%)	0.108	−0.209	0.161	−0.209	−0.152
	Weight (kg)	0.184	−0.225	−0.149	−0.006	−0.042
	BMI (kg/m^2^ )	0.200	−0.169	−0.084	0.16	0.127
	Back strength	−0.046	−0.078	−0.037	−0.279 *	−0.266 *
	Grip strength	0.014	0.071	0.139	−0.286 *	−0.211
	Repeat jump (count/30 s) 45 cm	−0.211	−0.222	−0.041	−0.241	−0.26
	Sit-up (count/60 s)	−0.032	0.181	0.119	−0.184	−0.122
	Sergeant Jump	−0.207	−0.158	−0.115	−0.430 **	−0.439 **
	Reaction time (sound)(1/1000 s)	0.237	0.117	0.097	0.333 *	0.349 **
	Side step (count/20 s)	0.036	0.118	−0.003	−0.15	−0.12
	Lung capacity: FVC (cc/kg)	−0.197	−0.165	0.113	−0.361 **	−0.323 *
	Resting heart rate (count/min)	0.052	−0.134	0.059	−0.292 *	−0.249
	20 m shuttle run (count)	−0.144	−0.23	0.049	−0.374 **	−0.350 **
	Sit-and-reach test (cm)	−0.022	−0.167	−0.158	−0.106	−0.147
	Single leg Stance (s)	−0.106	0.012	−0.018	0.165	0.132
Specific fitness factors	Bench press (kg)	0.056	0.109	−0.17	0.048	0.015
	Half squat (kg)	−0.021	0.014	−0.089	−0.358 **	−0.338 *
	Low body anaerobic (30 s)— Average power (W/kg)	0.143	−0.065	0.09	0.11	0.127
	Low body anaerobic (30 s)— Fatigue rate (%)	0.303 *	0.081	0.165	0.406 **	0.433 **
	Low body anaerobic (30 s)— maximum power (W/kg)	0.231	−0.024	0.14	0.296 *	0.315 *
	Low body anaerobic (30 s)— Average power (Watts)	0.217	−0.132	0.026	0.105	0.109
	Low body anaerobic (30 s)— maximum power (Watts)	0.261 *	−0.049	0.094	0.276 *	0.287 *
	Leg strength (60°/s)—Left and right average flexor (%BW)	−0.103	−0.307 *	−0.292 *	−0.448 **	−0.500 **
	Leg strength (60°/s)—Left and right average flexor (Nm)	−0.019	−0.327 *	−0.332 *	−0.377 **	−0.440 **
	Leg strength (60°/s)—Left and right average extensor (%BW)	−0.081	0.056	0.009	−0.241	−0.213
	Leg strength (60°/s) and right average extensor (Nm)	0.005	−0.001	−0.077	−0.192	−0.187
	GXT(Treadmill)—Drained time	0.025	−0.081	0.042	0.005	0.011
	GXT(Treadmill)—AT (%VO_2__max_)	−0.186	0.081	−0.159	0.062	0.005
	GXT(Treadmill)—AT (㎖/kg/min)	−0.136	0.006	−0.172	−0.007	−0.061
	GXT(Treadmill)—HR@AT (count/min)	0.106	−0.092	0.052	−0.012	0.005
	GXT(Treadmill)—HR_max_ (count/min)	0.21	−0.135	0.299 *	0.005	0.086
	GXT(Treadmill)—VE_max_ (ℓ/min)	−0.128	0.277 *	0.182	0.057	0.104
	GXT(Treadmill)—VO_2__max_ (㎖/kg/min)	0.015	−0.086	−0.079	−0.109	−0.12
	Maximal lactic acid concentration (mMol)	0.212	0.143	0.225	0.079	0.156
	Maximum leg strength ((180°/s)— average extensor AP (%BW)	0.041	0.074	0.015	−0.279 *	−0.232
	Maximum leg strength (180°/s)— average extensor AP (watts)	0.031	0.156	−0.012	0.152	0.146

* *p* < 0.05, ** *p* < 0.01, FS: fencing score, SS: swimming score, HS: horseback riding score, CS: combined score.

**Table 5 ijerph-18-04880-t005:** Comparison of national and international competitions in modern pentathlon: basic fitness factors and anthropometric data.

		Group	*N*	M	SD	*t*	df	*p*
Modern pentathlon	Fencing	International	57	213.73	14.76	−0.569	112	0.57
		National	57	215.30	14.66			
	Swimming	International	57	317.47	12.37	0.685	112	0.495
		National	57	315.75	14.37			
	Horseback riding	International	57	267.38	36.62	1.167	112	0.246
		National	57	258.01	48.28			
	Combined (shooting and running)	International	57	496.51	133.77	−4.549	112	0.001
		National	57	583.23	53.11			
	Overall performance	International	57	1295.09	151.99	−3.355	112	0.001
		National	57	1372.29	84.08			
Anthropometric data	Height	International	57	177.80	4.53	−0.125	112	0.900
		National	57	177.91	4.60			
	Body fat percentage	International	57	12.18	2.30	−0.09	112	0.929
		National	57	12.22	2.31			
	Weight	International	57	70.90	4.86	−0.084	112	0.933
		National	57	70.97	4.90			
	BMI	International	57	22.42	1.11	0.021	112	0.983
		National	57	22.41	1.09			
BFF	Back strength	International	57	138.10	15.39	0.142	112	0.887
		National	57	137.69	15.70			
	Grip strength	International	57	48.41	6.93	6.64	112	0.0001
		National	57	34.57	14.12			
	Repeat jump (count/30 s) 45 cm	International	57	28.62	8.72	−0.114	112	0.909
		National	57	28.81	8.80			
	Sit-up (count/60 s)	International	57	57.08	6.10	0.003	112	0.997
		National	57	57.08	6.10			
	Sergeant jump	International	57	56.18	9.52	0.039	112	0.969
		National	57	56.11	9.54			
	Reaction time (sound) (1/1000 s)	International	57	0.27	0.03	−0.005	112	0.996
		National	57	0.27	0.03			
	Side step (count/20 s)	International	57	44.59	2.63	−0.015	112	0.988
		National	57	44.60	2.63			
	Lung capacity: FVC(cc/kg)	International	57	84.48	6.37	0.316	112	0.752
		National	57	84.08	7.06			
	Resting heart rate (count/min)	International	57	66.10	6.85	0.015	112	0.988
		National	57	66.08	6.85			
	20 m shuttle run (count)	International	57	130.60	47.22	−0.031	112	0.975
		National	57	130.86	43.96			
	Sit-and-reach test (cm)	International	57	19.75	5.46	−0.115	112	0.909
		National	57	19.87	5.49			
	Single leg stance (s)	International	57	46.56	30.31	−0.081	112	0.936
		National	57	47.02	29.76			

BFF: Basic Fitness Factors.

**Table 6 ijerph-18-04880-t006:** Comparison of national and international averages of specific fitness factors.

Group			*N*	M	SD	*t*	df	*p*
SFF	Bench press (kg)	International	57	94.94	39.08	−0.203	112	0.839
		National	57	96.37	35.76			
	Half squat (kg)	International	57	147.86	33.79	−0.05	112	0.960
		National	57	148.16	31.27			
	Low body anaerobic (30 s)— average power (W/kg)	International	57	8.49	1.07	−0.001	112	0.999
		National	57	8.49	1.07			
	Low body anaerobic (30 s)— fatigue rate (%)	International	57	44.77	7.74	−0.25	112	0.803
		National	57	45.13	7.83			
	Low body anaerobic (30 s)—maximum power (W/kg)	International	57	11.70	1.91	−0.071	112	0.944
		National	57	11.72	1.92			
	Low body anaerobic (30 s)— average power (Watts)	International	57	599.61	90.28	−0.079	112	0.937
		National	57	600.94	90.50			
	Low body anaerobic (30 s)—maximum power (Watts)	International	57	824.77	156.25	−0.144	112	0.886
		National	57	829.01	158.31			
	Leg strength (60°/s)— Left and right average flexor (%BW)	International	57	170.49	21.91	0.16	112	0.873
		National	57	169.83	21.85			
	Leg strength (60°/s)— Left and right average flexor (Nm)	International	57	121.55	17.06	0.092	112	0.927
		National	57	121.26	17.05			
	Leg strength (60°/s)— Left and right average extensor (%BW)	International	57	303.18	32.05	7.171	112	0.0001
		National	57	262.53	28.37			
	Leg strength (60°/s)— and right average extensor (Nm)	International	57	215.89	23.71	0.067	112	0.947
		National	57	215.59	24.02			
	GXT(Treadmill)— time to exhaustion	International	57	20.53	0.92	0.062	112	0.951
		National	57	20.52	0.93			
	GXT(Treadmill)— AT (%VO_2__max_)	International	57	84.37	8.00	−0.103	112	0.918
		National	57	84.53	8.12			
	GXT(Treadmill)— AT (㎖/kg/min)	International	57	53.78	6.66	−0.093	112	0.26
		National	57	53.90	6.71			
	GXT(Treadmill)— HR@AT (count/min)	International	57	167.25	13.48	−0.101	112	0.920
		National	57	167.51	13.55			
	GXT(Treadmill) HR_max_ (count/min)	International	57	188.32	7.74	−0.068	112	0.946
		National	57	188.42	7.75			
	GXT(Treadmill)— VE_max_ (ℓ/min)	International	57	164.96	11.45	0.031	112	0.975
		National	57	164.90	11.45			
	GXT(Treadmill)— VO_2__max_ (㎖/kg/min)	International	57	63.71	4.71	−0.027	112	0.979
		National	57	63.73	4.70			
	Maximum lactic acid (mMol)	International	57	11.98	2.93	−0.074	112	0.941
		National	57	12.03	2.95			
	Maximum leg strength (180°/s)— average extensor AP(%BW)	International	57	377.25	55.54	3.597	112	0.0001
		National	57	328.60	85.70			
	Maximum leg strength (180°/s)— average extensor AP (watts)	International	57	259.49	36.10	1.795	112	0.075
		National	57	243.87	54.93			

SFF: specific fitness factors.

**Table 7 ijerph-18-04880-t007:** The effect of basic fitness factors on overall performance.

	Unstandardized Coefficient		*β*	*t*	*p*	*VIF*
	*B*	*SE*				
(Constant)	1449.7	323.3		4.484	0.000	
Sergeant Jump Grip Strength	−3.09 −2.19	1.919 0.987	−0.229 −0.223	−1.611 −2.219	0.11 0.029	2.699 1.356
Reaction time (sound) (1/1000 s)	−2.19	0.987	−0.223	−2.219	0.029	1.356
Side step (count/20 s)	483.7	405.4	0.124	1.193	0.236	1.456
Lung capacity: FVC (cc/kg)	−2.242	2.504	−0.117	−0.895	0.373	2.292
Side step (count/20 s)	3.953	5.235	0.081	0.755	0.452	1.531
Back strength	0.646	1.026	0.078	0.63	0.53	2.054
20 m shuttle run (count)	−0.208	0.365	−0.074	−0.569	0.57	2.243
Sit-and-reach test (cm)	−0.583	2.496	−0.025	−0.234	0.816	1.506
Sit-up (count/60 s)	−0.337	2.178	−0.016	−0.155	0.877	1.426
Single leg Standing (s)	0.006	0.434	0.001	0.014	0.989	1.372

** *F*(3.102) < *p* = 0.001 *R^2^*(*Adjusted R^2^*) = 0.231 (0.157) *VIF* < 10.0.

**Table 8 ijerph-18-04880-t008:** The effect of specific fitness factors on overall performance.

	Unstandardized Coefficient		*β*	*t*	*p*	*VIF*
	*B*	*SE*				
(Constant)	1259.3	355.9		3.538	0.001	
Bench press(kg)	1.222	0.386	0.355	3.161	0.002	1.908
Half squat(kg)	−1.097	0.514	−0.277	−2.136	0.035	2.548
Low body anaerobic (30 s)—average power (W/kg)	14.077	13.55	0.116	1.039	0.301	1.900
Leg strength (60°/s)— left and right average flexor (%BW)	−2.338	0.585	−0.397	−3.995	0.001	1.493
GXT(Treadmill)—drained time	24.712	14.35	0.177	1.722	0.088	1.605
Maximum lactic acid concentration (mMol)	2.471	4.265	0.056	0.579	0.564	1.433
Maximum leg strength (180°/s)— average extensor AP (%BW)	−0.393	0.159	−0.232	−2.462	0.015	1.347

** *F*(6.451) < *p* = 0.001, *R^2^*(*Adjusted R^2^*) = 0.299, (0.252) *VIF* < 10.0.

## Data Availability

Please refer to suggested Data Availability Statements in section “MDPI Research Data Policies” at https://www.mdpi.com/ethics.

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
