# Peer review of "Relative Contribution among Physical Fitness Factors Contributing to the Performance of Modern Pentathlon"

_ijerph, 2021, doi:10.3390/ijerph18094880_

Round 1

Reviewer 1 Report

Dear Authors!

The manuscript I received entitled "Relative contribution among physical fitness factors contributing to the performance of modern pentathlon" raises my emotions, because the issues related to the search for methods and tools optimizing and rationalizing the effectiveness of the training process are very close to me.

Congratulations on taking up the topic, congratulations on the development of the material!

However, the material sent to me for review, at the moment, does not meet the requirements for scientific studies and requires many corrections.

Arguments:

  • Lack of worldwide literature. The literature review is limited to Korean authors and Korean publishers only.
  • The description of the research subject is limited to giving gender and number of participants only. Insufficient characterization for a potential reader.
  • The biggest drawback is the lack of access to research results, which in my opinion should be visible. The link in the text to the UIPM website (http: ///www.uipmworld.org) does not work. Enclosed is a screenshot.
  • For this reason, further questions arise. What criteria were decisive for the selection to the research group (n = 57). What determined the selection of individual parameters / physical fitness tests (lack of clarity)? According to what methodology were they selected for analysis? What is meant by basic and specific physical fitness in the modern pentathlon? Where is it explained?
  • Lack of detailed description of the methodology and procedures of statistical analyzes.
  • It should be remembered that the individual contests in the modern pentathlon are very different, whether due to the environment (running, swimming) or due to the multitude of factors determining the final result (horse, opponent).
  • There are numerous typing and editorial errors in the Results chapter.
  • Tables are hard to read.
  • In many cases, the tables do not provide the measurement unit for a given parameter.
  • The names of the analyzed parameters given in the tables are often incomprehensible (their description / legend is missing).
  • Often the Authors use the so-called mental shortcuts.
  • The conclusions presented at the end do not fully correspond to the previously formulated aims of the work, e.g. “This study can provide valid evidences for fitness assessment and testing selection, and can also be utilized for pentathlete selection process, fitness assessment, training distribution and planning as it includes objective investigation on the pentathlon performance factors ”.
  • A final reflection on this issue should be formulated.

Once again, I congratulate the Authors on taking up this research problem. I hope you will find my comments helpful.

Good luck.

Author Response

Thank you for your careful review.
I organized the contents in the attached below.

Comments and Suggestions for Authors

Arguments:

Lack of worldwide literature. The literature review is limited to Korean authors and Korean publishers only.

-> A worldwide literature paper on modern pentathlon was added.

The description of the research subject is limited to giving gender and number of participants only. Insufficient characterization for a potential reader.

-> Thank you for your careful review. It is desirable to give readers good information for follow-up research to explain the characteristics of the subjects in detail. However, please understand that the main subjects of this study are based on the analysis results of the Korean national team's modern pentathlon, so we cannot present them in detail.

The biggest drawback is the lack of access to research results, which in my opinion should be visible.

The link in the text to the UIPM website (https://www.uipmworld.org) does not work. Enclosed is a screenshot.

-> I modified the wrong web address to make it accessible.

-> And the screenshot attached below is a website that holds data for national athletes of the Korea Institute of Sports Science, which is not accessible unless you are a researcher in charge.

For this reason, further questions arise. What criteria were decisive for the selection to the research group (n = 57). What determined the selection of individual parameters / physical fitness tests (lack of clarity)? According to what methodology were they selected for analysis? What is meant by basic and specific physical fitness in the modern pentathlon? Where is it explained?

-> The study subjects used as the result analysis data of this study are the data of Korean modern pentathlon national team players. The 57 cases are explained in this study, but the number of cases is small, but it is a population because it is a measure of 57 national players from 2005 to 2019.

Lack of detailed description of the methodology and procedures of statistical analyzes.

-> The researchers believe that the descriptions of measurement variables, data collection methods, and data analysis procedures presented in the research method have applied and described appropriate methods to achieve the purpose of this study. Please understand.

-> It should be remembered that the individual contests in the modern pentathlon are very different, whether due to the environment (running, swimming) or due to the multitude of factors determining the final result (horse, opponent).

-> I’m very much agree with the reviewer. Modern pentathlon is a sport in which 5 different sports are performed in different stadium environments.

There are numerous typing and editorial errors in the Results chapter.

-> Corrected typing errors and edits.

Tables are hard to read.

-> As a researcher, I partially agree with the reviewer that it is difficult to see the results table. However, as we try to put as many analytical variables as possible in one table, it may be difficult to see. Please understand.

In many cases, the tables do not provide the measurement unit for a given parameter.

-> Modified and supplemented.

The names of the analyzed parameters given in the tables are often incomprehensible (their description / legend is missing).

-> Modified and supplemented.

Often the Authors use the so-called mental shortcuts.

->

The conclusions presented at the end do not fully correspond to the previously formulated aims of the work, e.g. “This study can provide valid evidences for fitness assessment and testing selection, and can also be utilized for pentathlete selection process, fitness assessment, training distribution and planning as it includes objective investigation on the pentathlon performance factors ”.

-> Modified and supplemented.

A final reflection on this issue should be formulated.

Once again, I congratulate the Authors on taking up this research problem. I hope you will find my comments helpful.

Good luck.

Reviewer 2 Report

In this study we talk about competition records, but not about the number of subjects.  It is necessary to know how many different subjects have been recorded.
On the other hand, if there are several records of the same subject, the relationships between the measurements of these records, as well as their evolution, must be studied.
Please include these data in the Material and Methods section and in the Results and Conclusions section.

Author Response

Thank you for your careful review.
I organized the contents in the attached below.

Comments and Suggestions for Authors

In this study we talk about competition records, but not about the number of subjects.

-> Modified and supplemented.

It is necessary to know how many different subjects have been recorded.

On the other hand, if there are several records of the same subject, the relationships between the measurements of these records, as well as their evolution, must be studied.

-> A player may have a record of five events. However, in modern pentathlon, ranking is selected by overall score, so we analyzed it with the final ranking rather than analyzing it by the scores of each record and physical strength.

Please include these data in the Material and Methods section and in the Results and Conclusions section.

-> Modified and supplemented.

Round 2

Reviewer 2 Report

Thank you to the authors for responding to the comments submitted.